# Plasma Exfoliated Graphene: Preparation via Rapid, Mild Thermal Reduction of Graphene Oxide and Application in Lithium Batteries

**DOI:** 10.3390/ma12050707

**Published:** 2019-02-28

**Authors:** Zuyun Luo, Yuanyuan Li, Fangfang Wang, Ruoyu Hong

**Affiliations:** 1College of Chemical Engineering, Fuzhou University, Fuzhou 350116, China; fzu_lzy@163.com (Z.L.); N160427010@fzu.edu.cn (Y.L.); wangff19900625@163.com (F.W.); 2College of Zhicheng, Fuzhou University, Fuzhou 350002, China

**Keywords:** graphene oxide, low-temperature plasma, cathode material, lithium battery

## Abstract

A simple, novel approach is proposed for the preparation of plasma-exfoliated graphene (PEGN) by reducing graphene oxide (GO) through a dielectric-barrier discharge (DBD) plasma treatment in a H_2_ atmosphere. The surface chemistry, microstructures, and crystallinity of the prepared samples were characterized via X-ray photoelectron spectroscopy, transmission electron microscopy, and Raman spectrometry to determine the formation mechanism of the PEGN. The results demonstrated that the prepared PEGN had only a few layers in its structure and that most of the functional groups containing oxygen on the GO surface were removed. The PEGN exhibited a considerably higher capacity, better cycling stability, and favorable electron transfer rate for use as a cathode material for lithium-ion batteries. This proposed approach is fast, convenient, and inexpensive, constituting a novel means of producing graphene.

## 1. Introduction

Graphene, a flat monolayer of carbon atoms that are packed into 2-D honeycomb lattices, has attracted considerable attention in recent years for its high thermal and electrical conductivity, outstanding mechanical strength, and large specific surface area. Graphene-based materials have the potential for application to several fields, such as energy storage [1,2], transparent electrodes [3], lithium-ion batteries (LIBs) [4,5,6,7], solar cells [8], fuel cells [9,10], air separation [11], and building materials [12]. Therefore, there is a need for high-quality, large-scale graphene samples. To date, various methods have been developed for fabricating graphene, such as the exfoliation of highly oriented pyrolytic graphite via mechanical methods [13], sublimating silicon from silicon carbide [14,15], thermal- [16,17] or plasma-strengthened chemical vapor deposition [18], and the thermal exfoliation and reduction of graphene oxide (GO) [19]. Therein, synthesizing large quantities of GO from inexpensive graphite powder is a potential alternative solution for the batch manufacturing of materials based on graphene. Nevertheless, the chemical reduction of GO requires the use of strong chemical reductants, such as hydrazine (N_2_H_4_) [20,21] and sodium borohydride (NaBH_4_) [22], both of which are dangerous and environmental pollutants. Alternatively, removing oxygen in an Ar atmosphere, a H_2_ atmosphere, or an ultrahigh-vacuum environment through thermal annealing has also been proposed [23]; however, the high-temperature requirement restricts the range of application of such substrates. Flash reduction [24] and electrochemical reduction [25] are novel, environmentally friendly, and low-temperature methods of GO reduction. With these methods, plasma discharge can make the effective removal of oxygen-containing functional groups possible during the low-temperature generation of high-purity atomic hydrogen [26]. However, there are energetic species (such as ions) in plasmas and these energetic species can sputter or damage the materials of atomically thin layers [27].

Therefore, the development of new reduction techniques involving non-toxic chemicals is necessary for the rapid fabrication of large batches of high-quality graphene. The present study addressed a simple, low-cost, and environmentally friendly approach to the synthesis of plasma-exfoliated graphene (PEGN) by reducing GO with dielectric barrier discharge (DBD) plasma under atmospheric pressure. Cyclic voltammetry (CV) measurements revealed that the PEGN, when applied as an anode material for LIBs, exhibited a significantly high rate capability with excellent cycling stability and a high capacitance. 

## 2. Materials and Methods 

### 2.1. Reagents and Materials

Natural graphite flakes (approximately 300 mesh) were purchased from Qingdao Risheng Co., Ltd. (Qingdao, China). Analytically pure hydrochloric acid was obtained from Quanzhou Donghai Chemical Reagents Company (Quanzhou, China). Sodium nitrate was provided by Taicang Chemical Reagents Co., Ltd. (Taicang, China). Barium chloride and 30% H_2_O_2_ aqueous solution were purchased from Shanghai Wokai Biotechnology Co., Ltd. (Shanghai, China). Potassium permanganate and 98% H_2_SO_4_ were supplied by Shanghai Chemical Reagents Company (Shanghai, China). The LiFePO_4_ (industrial purity) and N-methyl-2-pyrrolidone (battery level) were supplied by Sinopharm Chemical Reagent Co., Ltd. (Shanghai, China). The carbon black and binder (industrial purity) were obtained from Decatur Battery Technology Co., Ltd. (Shenzhen, China). The lithium foil and electrolyte solution (battery level) came from Dongguan Shanshan Battery Materials Co., Ltd. (Dongguan, China). All the chemicals were used directly without further purification. Ultra-pure water was produced using a Millipore system (Burlington, MA, USA).

### 2.2. Preparation of GO

GO was prepared from graphite flakes using a modified Hummers method [28,29]. The reaction took place at three temperatures: First, 250 g of natural graphite powder was added to a 50 L glass reactor followed by 125 g of NaNO_3_ and 5.75 L of H_2_SO_4_ that were stirred for 15 min over an ice-water bath. Next, 750 g of KMnO_4_ was poured into the mixture in batches. The mixture was allowed to continue to react under vigorous agitation for 2 h. Then, the reaction entered the mesothermal stage. The ice-water bath was removed 5 min later, after which the system was heated to 35 °C and stirred for 4 h. This led to the start of the high-temperature reaction stage in which 12.5 L of water was slowly poured into the solution. The reaction was then sustained for an additional 30 min. This was followed by the addition of a 3 wt% H_2_O_2_ aqueous solution until no further bubbling was observed. Finally, 5 L of 5 wt% dilute hydrochloric acid was added. Once the reaction was complete, the system was low speed centrifuge washed twice, until the pH reached about 3. The sample was then transferred to a hollow fiber membrane for washing to neutrality. After the trace black residue had been filtered out, the homogeneous suspension was collected. The brownish-yellow GO was obtained after drying the suspension using a spray-drying method.

### 2.3. Preparation of the PEGN with DBD Plasma

Figure 1a is a schematic diagram of the apparatus used for the synthesis of PEGN at ambient pressure. GO powder was placed on a porous plate at the center of a vertical quartz tube in a reducer. Prior to discharge, to purify the air in the tube, inert Ar gas was passed through the fluidized bed for 10 min at 100 mL min^−1^. Then, the DBD plasmas were initiated with an AC input voltage of 50 V and a current of 1.2 A at ambient laboratory temperature. H_2_ gas was introduced to the plasma to generate hydrogen plasma. The stripped product was then separated using a cyclone separator, ultimately producing black PEGN powder. The significant expansion of the GO powder in the discharge process at different times is shown in Figure 1b. Figure 1c shows the discharge process.

### 2.4. Characterization of PEGN 

A Hitachi S-4800 scanning-electron microscope (SEM, Tokyo, Japan) and Hitachi H-600-II transmission electron microscope (TEM) were used to characterize the morphology of the PEGN nanostructures. The critical structures in the PEGN particles were recorded using a Siemens D8 Advance X-ray diffractometer (XRD, Munich, Germany) with Cu-Ka radiation (=0.15418 nm) over a range of 5° ≤ 2° ≤ 80°. The Raman analysis was carried out using a Via-Reflex Raman microspectrometer (Renishaw, Wotton-under-Edge, UK) using a continuous-wave laser with a wavelength of 532 nm. Under carbon-hybridized conditions, surface chemical statements were recorded using an X-ray photoelectron spectroscope (XPS, Escallab 250, Thermo Scientific, Waltham, MA, USA) with an Al-Kα line and a VG CLAMP hemispherical analyzer (Waltham, MA, USA). 

### 2.5. Electrochemical Measurement of PEGN 

The prepared PEGN was used to fabricate a coin-type half-cell. This was used as a cathode conductive additive for a LIB with which the electrochemical performance of the PEGN was evaluated. Working electrodes were produced through the mixing of LiFePO_4_, PEGN, carbon black (CB), and a poly(vinylidene fluoride) binder that was dispersed in a N-methyl-2-pyrrolidone solution. When 10 wt% CB was used as a conductive additive, the amount of binder was 10 wt%. When using PEGN 3 wt% and a composite conductive agent of 2 wt% PEGN and 1 wt% CB, the binder was added in an amount of 3 wt%. Before assembling the coin cells, we spread a prepared slurry containing the active materials uniformly onto the aluminum foil and then dried it in a vacuum oven for 12 h at 110 °C. LIR2025-type coin cells, assembled in a Lab2000 dry glove box filled with argon (Etelux, Beijing, China), were employed in the subsequent electrochemical experiments. In addition, LiFePO_4_@PEGN/CB or LiFePO_4_@CB, pure lithium foil, polypropylene membrane film, and 1 M LiPF_6_ in ethylene carbon (EC)-dimethyl carbonate (DMC) were used as the working electrode, counter and reference electrodes, separator, and electrolyte, respectively. The galvanostatic charge-discharge (GCD) behavior was measured on a Neware GCD system (CHI600E, Shanghai, China). A CHI-760E electrochemical analyzer (Huakeputian technology Co., Ltd, Beijing, China) was used for CV at a scanning rate of 0.1 mV/s at voltages in a range of 2.6 to 4.3 V (v. Li^+^/Li). The CHI-760E was used for electrochemical impedance spectroscopy (EIS). All electrochemical measurements were carried out at ambient laboratory temperatures.

## 3. Results and Discussion

### 3.1. Characterization Studies

The schematic representation of the exfoliation and reduction of GO by DBD plasma technology is shown in Figure 2a. During the treatment, the plasma enabled burst open, high-energy collisions between the electrons and ions which were used for exfoliation. During DBD plasma treatment, H_2_O and CO_2_ gases were discharged abruptly to exfoliate the GO to a thickness of several few-layers. As shown in Figure 2a, 200 mg of GO only covered a small area of a watch glass. However, after treating the GO with H_2_ DBD plasma for 3 min, the majority of the surface of the watch glass was covered with the reaction product. This volumetric expansion indicated that the layers of the GO were efficiently and significantly exfoliated. Then, the samples were characterized using SEM, TEM, and atomic force microscopy (AFM, Agilent 5500, NYSE:A, PaloAlto, Santa Clara, CA, USA) to ascertain the structure and thickness of the products. The SEM images in Figure 2c show that thin sheets were produced through the use of the H_2_ plasma compare with GO as show in Figure 2b. A typical TEM image (Figure 2d) shows that there are some wrinkled or folded regions on the planes of the sheet structures. This was probably a consequence of the minimal thickness of the sheets. The presence of the folded structures at the edges of the PEGN, as seen in the HRTEM image (Figure 2e), indicates that most of the PEGN is few-layer. The corresponding selected area electron diffraction (SAED) is shown in Figure 2f. The SAED pattern was represented by weak diffraction rings and diffraction spots, which indicated the loss of the long-range ordering in the graphene layers and the well crystallized few-layer graphene structure [30,31,32]. Figure 2g is a morphology map of graphene as captured by AFM. The corresponding thickness information is shown in Figure 2h. The thickness of the graphene was about 0.4–0.5 nm, which was slightly greater than the theoretical thickness of single-layer graphene (0.34 nm). This difference was caused by the uneven surface of the graphene and the presence of a mica sheet at the base. We concluded that, by applying a low-temperature H_2_ plasma technique, few-layers PEGN can be fabricated from GO.

Then, we further characterized the GO and PEGN from the N_2_ adsorption and desorption isotherms (3Flex, MicroMetric Inc., Sarasota, FL, USA) at 77 K. It was observed that the S_BET_ value of the PEGN was 475 m^2^/g, while that of the GO was 101.5 m^2^/g. The larger specific surface area was advantageous to the formation of a conductive network. Figure 3a shows that the isotherms were of type-IV with H_2_ hysteresis loops at relatively high pressures. This indicated that these materials consist of aggregated sheets in which there are typical mesoporous microstructures, while the pore size was found to be approximately 2.38 nm (inset, Figure 3a). The Raman spectra of GO, graphite, and PEGN are shown in Figure 3b. These three spectra all exhibit a D and G band. The G band of PEGN shifted to 1591 cm^−1^ while that of GO was at 1598 cm^−1^, approximating to the value of pristine graphite, which suggests that the GO was reduced through the plasma treatment. The D band for both GO and PEGN overtopped that of pristine graphite, predicting the presence of defects in the sample and the size of the in-plane sp^2^ domain. The intensity ratio ID/IG of the D and G bands varied from 0.94 to 0.95, suggesting a slight increase in the average size of the sp^2^ domain after the plasma-treatment-induced reduction of GO [33]. A high energy second-order 2-D peak of the prepared PEGN appeared at around 2656 cm^−1^ (Figure 3b). This was in good agreement with the results obtained from the examination of a few layers of graphene, fabricated by micromechanical cleavage [34], and chemically-reduced GO [35].

FT-IR, XPS, and XRD investigations were conducted to explore the structural characteristics and chemical composition of GO, graphite, and the product of the plasma treatment. Figure 4 shows the FT-IR spectra of GO, the products of the plasma treatment after 1, 2, and 3 min, and natural graphite. The characteristic peaks of GO appeared at 1730, 1622, 1414, 1228, and 1116 cm^−1^, corresponding to the stretching oscillation of C=O, aromatic C=C, carboxy C–O, epoxy C–O, and C–O, respectively [36]. The peaks of oxygen-containing functional groups gradually decreased in amplitude and finally the peak of carboxy C–O at 1414 cm^−1^ disappeared completely after the plasma treatment. Furthermore, the intensity of the peaks corresponding to aromatic C=C at 1622 cm^−1^ increased with increasing the time of the plasma treatment, primarily because the graphite structure was recovered with the increase of the reduction degree.

Layer spacing is a significant parameter affecting the evaluation of the structural information of graphene. Figure 5 shows the XRD spectra of GO and the products obtained through plasma treatment for different time periods (1, 2, and 3 min). Compared to natural graphite, GO had a broad peak (about 10°), which implied that the regular multilayer crystalline texture of graphite was broken down as a result of oxidation. To investigate the effects of the chosen preparation conditions of PEGN through DBD plasma, we carried out comparative research using different working times. For the product fabricated by plasma-treating GO for 3 and 2 min, no peak was observed, thereby suggesting that the multilayer structure was lost and that single-layer graphene was formed [33].

The results of XPS measurements provided direct evidence for the reduction of GO during plasma treatment. The XPS patterns of GO and the PEGN (obtained via plasma treatment for 3 min) are shown in Figure 6. Figure 6a shows the results of a survey scan of the PEGN compared to that of GO. A C 1s peak appears at about 284 eV, simultaneously with the O 1s peak at about 530 eV. Figure 6b,c shows the C1s high-resolution XPS spectra of GO and PEGN. For GO, the C1s spectrum was fitted to the four components at 284.6 eV (C–C), 286.1 eV (C–O), 287.5 eV (C=O), and 289.2 eV (O–C=O); further, the amplitudes of the peaks at 286.1 eV and 287.5 eV decreased after the plasma treatments, and the peak at 289.2 eV disappeared. This is in good agreement with Figure 6a which indicates that the O 1s peak of GO is lower than that of the PEGN. This indicated that oxygen was removed from the GO to form graphene [23,37].

### 3.2. Electrochemical Properties

To determine whether PEGN could be used to build effective conducting networks for both the ions and electrons that must be transported by the cathode system of LIBs, the electrochemical impedance spectroscopy (EIS), and cyclic voltammetry (CV) measurements were performed on the LiFePO_4_@CB, LiFePO_4_@PEGN, and LiFePO_4_@PEGN/CB electrodes before attempting to evaluate their charge/discharge cycle performances. Figure 7a shows the CV profiles for three different electrodes, scanned at a rate of 0.1 mV/s. As shown in Figure 7a, the redox peaks of the LiFePO_4_@CB, LiFePO_4_@PEGN, and LiFePO_4_@PEGN/CB electrodes are located at 3.270/3.573, 3.278/3.618 and 3.337/3.518 V, respectively. A larger current peak and narrower peak potential separation were found for the LiFePO_4_@PEGN/CB, while the peak potential difference for LiFePO_4_@PEGN/CB (181 mV) was significantly less than that of LiFePO_4_@PEGN and LiFePO_4_@CB (303 mV). This suggests that the addition of PEGN made the electrode reaction more reversible and enhanced the kinetic behavior of these electrodes.

Figure 7b shows the EIS results for LiFePO_4_@CB, LiFePO_4_@PEGN, and LiFePO_4_@PEGN/CB. They are linear and have a depressed semicircular shape in the low- and high-frequency regions of the Nyquist spectra, respectively. The semicircular shape is associated with the resistance to the electrolyte and charge transfer, while the straight line is related to the constant phase element and ion diffusion within the anode. As shown in Figure 7b, the radius of LiFePO_4_@PEGN/CB (79 Ω) was smaller than that of LiFePO_4_@PEGN (106 Ω) and LiFePO_4_@CB (150 Ω), indicating that as the resistance decreased, the rate of electron transfer and electrical conductivity increased. We can speculate that adding PEGN improved the conductivity, promoted electron transfer, and thus reduced the resistance. Moreover, the plots for LiFePO_4_@PEGN/CB, LiFePO_4_@PEGN, and LiFePO_4_@CB have similar shapes, with an arc and an inclined line in the high- and low-frequency regions. The former was generally attributed to resistance resulting from the charge transfer between the active material and liquid electrolyte, while the latter appears as Li^+^ diffused into the electrode materials (the so-called Warburg diffusion effect [38]), as a result of ion diffusion. The ions were then transported to the electrode surface depending on the frequency [39].

The electrochemical performances of the three types of LiFePO_4_ electrodes are compared in Figure 8. Figure 8a shows the curves of the initial charge and discharge of electrodes prepared using 2 wt% PEGN plus 1 wt% carbon black, 3 wt% PEGN or 10 wt% carbon black additive at a current density of 20 mA/g. Although the PEGN electrode contained a much lower proportion of additives, the LiFePO_4_@PEGN/CB had a higher initial discharge specific capacities of approximately 161 mAh/g than LiFePO_4_@PEGN for 156 mAh/g and LiFePO_4_@CB for 146 mAh/g, respectively. The capacity of the PEGN electrode was significantly higher than that of an electrode fabricated with a commercial conductive additive. Therefore, we can conclude that a much more effective conductive network can be built in a LIB cathode by adding 2 wt% PEGN plus 1 wt% carbon black, thus improving its capacity. Figure 8b shows the cycling performances of the three types of LiFePO_4_ electrodes. The discharge specific capacity of LiFePO_4_@PEGN/CB remained at 152 mAh/g even after 100 cycles that were in accord with Su [40]. Furthermore, the coulombic efficiencies of the LiFePO_4_@PEGN/CB, LiFePO_4_@PEGN, and LiFePO_4_@CB were 94.4%, 93.8%, and 90.4%, respectively, after 100 cycles. These results are a further indication that the addition of PEGN improved the capacity and conductivity. The main reason is that there is a good complementary effect in the contact modes of the graphene and carbon black when using the composite conductive agent. It can establish a more effective “long-range” and “short-range” conductive network inside the electrode [41,42]. This binary conductive agent can improve the dispersion of graphene and prevent the aggregation of graphene, that further improves the efficiency of the electrical conduction [43].

## 4. Conclusions

We propose a simple approach for the preparation of graphene by exfoliating and reducing GO with DBD plasma in an H_2_ atmosphere. As a result, most of the functional groups containing oxygen were removed from the surface of the GO. The resulting graphene exhibited a much higher capacity, as well as superior cycling stability and rate performance, compared to carbon black when used as the cathode material in an LIB. This approach offers a simple, environmentally friendly, and cost-effective approach that can be easily applied to the mass production of graphene materials under moderate conditions.

## Figures and Tables

**Figure 1 materials-12-00707-f001:**
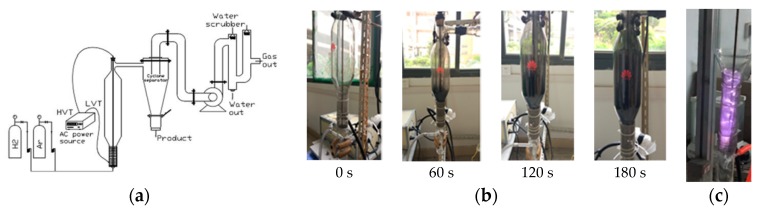
(**a**) Schematic representation of the apparatus used to reduce graphene oxide (GO). Low voltage terminal (LVT) and high voltage terminal (HVT) are connected to the low- and high-voltage terminals, respectively; (**b**) expansion of GO powders after treatment with dielectric-barrier discharge (DBD) plasma for 0, 60, 120, and 180 s, respectively; (**c**) discharge process.

**Figure 2 materials-12-00707-f002:**
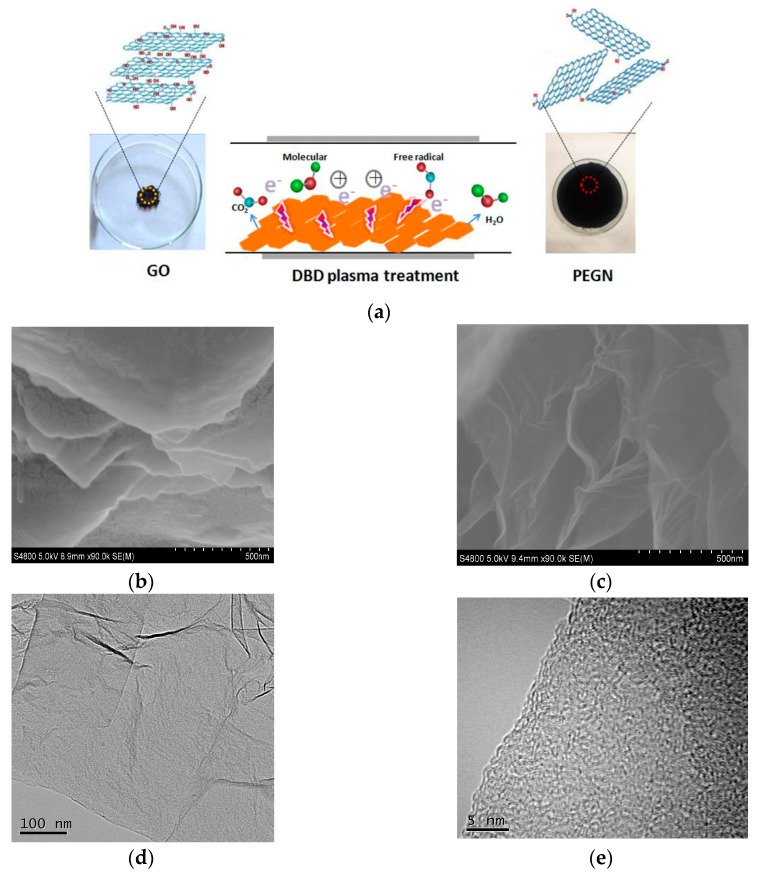
(**a**) Schematic representation of exfoliation and reduction of GO using DBD plasma technology; SEM images of (**b**) GO and (**c**) plasma-exfoliated graphene (PEGN); (**d**) TEM and (**e**) HRTEM images of PEGN; (**f**) selected area electron diffraction (SAED) pattern indicating a disruption of long-range ordering in graphene; (**g**) atomic force microscopy (AFM) images of PEGN (scale bar = 500 nm). (**h**) Height profiles along line in (**f**).

**Figure 3 materials-12-00707-f003:**
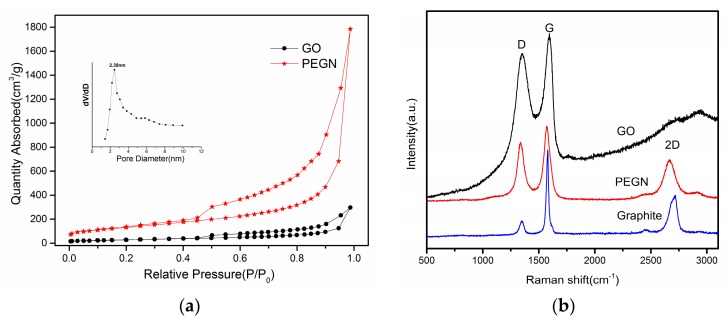
(**a**) Nitrogen adsorption and desorption isotherms of PEGN fabricated using H_2_ DBD plasma at 77 K (inset: distribution of pore sizes). (**b**) Raman spectra of graphite, GO, and PEGN.

**Figure 4 materials-12-00707-f004:**
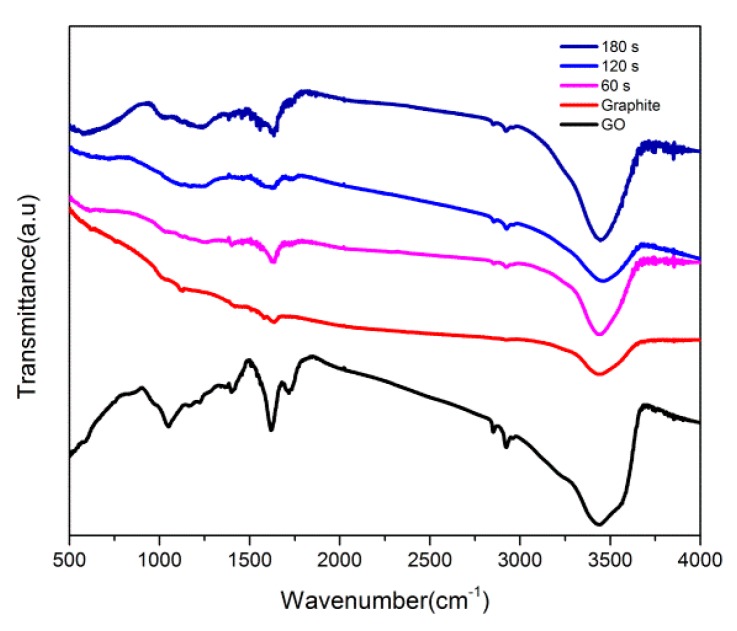
FT-IR spectra of GO, graphite, and the products of plasma treatment for 60, 120, and 180 s, respectively.

**Figure 5 materials-12-00707-f005:**
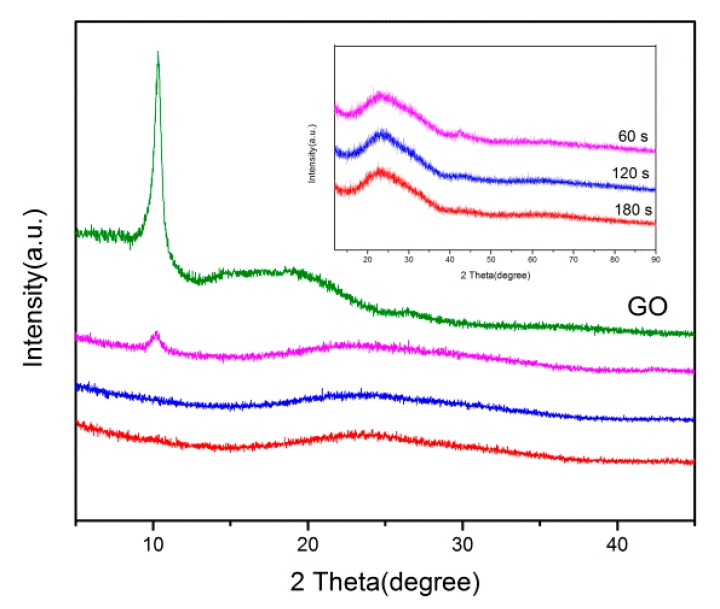
XRD patterns of GO and products of plasma treatment for 1, 2, and 3 min.

**Figure 6 materials-12-00707-f006:**
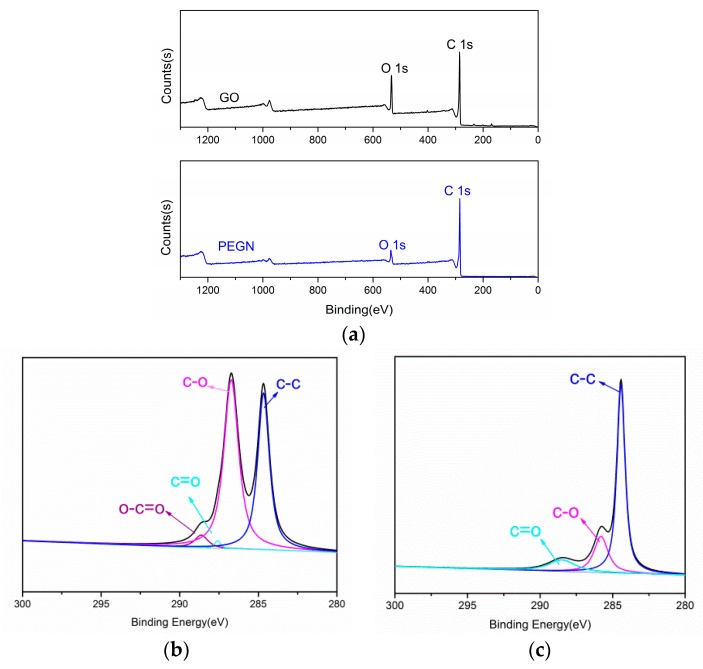
XPS spectra of GO and PEGN. (**a**) Survey scan of GO and PEGN. High-resolution C1s spectra of (**b**) GO and (**c**) PEGN.

**Figure 7 materials-12-00707-f007:**
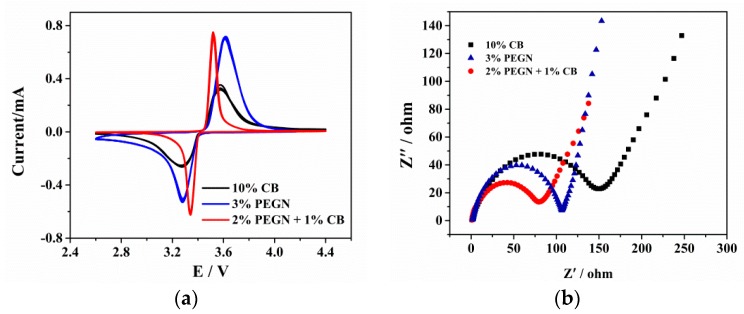
(**a**) Cyclic voltammetry (CV) profiles and (**b**) electrochemical impedance spectroscopy (EIS) patterns for LiFePO_4_@PEGN/CB, LiFePO_4_@PEGN, and LiFePO_4_@CB.

**Figure 8 materials-12-00707-f008:**
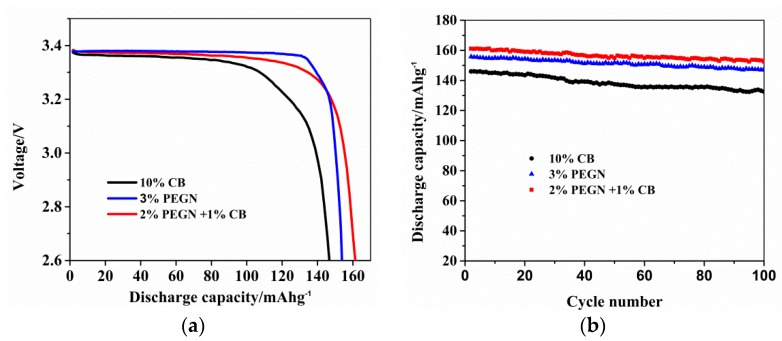
(**a**) Discharge curves and (**b**) cycle performance of LiFePO_4_@PEGN/CB, LiFePO_4_@PEGN, and LiFePO_4_@CB.

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
