# Peer review of "Plasma Exfoliated Graphene: Preparation via Rapid, Mild Thermal Reduction of Graphene Oxide and Application in Lithium Batteries"

_materials, 2019, doi:10.3390/ma12050707_

Reviewer 1 Report

The submitted manuscript reports the preparation, the structural characterization and the evaluation of the electrochemical properties of a plasma exfoliated graphene through reduction of Graphene Oxide (GO). The subject is topical and many research efforts have been made up to now in order to overcome the barriers of the current Li-ion technology.

The matter is satisfactorily presented, and the structural characterization is comprehensive. Though, the author should provide more information about the graphene nanosheets thickness values, since it is known that single-layer graphene is not chemically stable and tends to aggregate in few-layer stacks.

The electrochemical tests of this product are limited only to the LiFePO4 cathode, which does not meet the requirements as high capacity material for the current required energy density increase in Li-ion technology. The binder quote in the electrode preparation procedure is missing.  Its application only as conductive additive limits the disclosure of the actual performance of the graphene material and should also be compared with an industry available graphene.

In general, the manuscript lacks a certain degree of novelty. And since the preparation procedure, from GO to the final Reduced Graphene Oxide is not easily scalable, with the high costs associated to graphene production, an industrial scale-up of graphene to be used as conductive additive is likely to be poorly both environmentally and economically sustainable.

For all these reasons the manuscript is not suitable for publication in Materials in the current state.

Author Response

Dear Reviewers: 

Thank you for your reviewers’ comments concerning our manuscript entitled “Plasma exfoliated graphene: preparation via rapid, mild thermal reduction of graphene oxide and application in lithium batteries” (ID: materials-424271). Those comments are all valuable and very helpful for revising and improving our paper, as well as the important guiding significance to our researches. We have studied comments carefully and have made correction which we hope meet with approval. Revised portion are marked in red in the paper.

The responds to the reviewer’s comments are as flowing:

Point 1: the author should provide more information about the graphene nanosheets thickness values, since it is known that single-layer graphene is not chemically stable and tends to aggregate in few-layer stacks.

Response 1: Thanks for the reviewer’s kind advice. Here, The graphene nanosheets thickness were characterized by a variety of methods, for instance TEM、HRTEM、AFM、and Raman. The corresponding thickness information of the graphene shown in Figure 2g, is about 0.4–0.5 nm, which is slightly greater than the theoretical thickness of single-layer graphene (0.34 nm). This difference is caused by the uneven surface of the graphene and the presence of a mica sheet at the base. Through Raman analysis as show in Figure 3b, A high-energy second-order 2-D peak of the prepared PEGN appeared at around 2656 cm-1.This was in good agreement with the results obtained from the examination of a single layer of graphene, fabricated by micromechanical cleavage and chemically reduced GO. 

Point 2: The electrochemical tests of this product are limited only to the LiFePO4 cathode, which does not meet the requirements as high capacity material for the current required energy density increase in Li-ion technology.

Response 2: Our study is trying to demonstrate whether the graphene we prepared is able to construct an effective conducting network for both electron and ion transports in cathode system of a high-power lithium ion battery (LIB) based on a coin cell. In the experiment, we choose LiFePO4 as an cathode material for lithium ion batteries. The discharge specific capacity of LiFePO4@PEGN/CB remained at 152 mAh/g even after 100 cycles (the theoretical specific capacity of LiFePO4 is 170 mAh×g-1). As show in Figure 7 and Figure 8, the CV profiles、the EIS and the electrochemical performances of the addition of graphene in cathode could effectively improve the capacity and electron and ion transports.

 Point 3: The binder quotein the electrode preparation procedure is missing.

Response 3: We are very sorry for our negligence. In our study, the binder quotein is poly(vinylidene fluoride),We have supplied according to the Reviewer’s comments and marked in red.

Point 4: Its application only as conductive additive limits the disclosure of the actual performance of the graphene material and should also be compared with an industry available graphene.

Response 4: We are very sorry for no comparative analysis was performed with commercial graphene. But the cycle characterizations of the LiFePO4@PEGN/CB  is quite stable as well as previous study even after 100 cycles. And as Reviewer suggested that We will expand the application research of graphene.

Point 5:In general, the manuscript lacks a certain degree of novelty. And since the preparation procedure, from GO to the final Reduced Graphene Oxide is not easily scalable, with the high costs associated to graphene production, an industrial scale-up of graphene to be used as conductive additive is likely to be poorly both environmentally and economically sustainable.

Response 5: Up to now, various methods have been developed for fabricating graphene, but large-scale preparation of high-quality graphene still faces many difficulties.Through reduction of graphene oxide to prepare graphene is one of the most important ways at low cost and on a large scale. So, a simple, novel approach is proposed for the preparation of graphene through a DBD plasma treatment in a H2 atmosphere that efficiently and environmentally. We are also further studying the expansion of the reactor to improve production efficiency through the technique of combining plasma with a fluidized bed based on the current research.

      We tried our best to improve the manuscript and made some changes in the manuscript.  These changes will not influence the content and framework of the paper. And here we did not list the changes but marked in red in revised paper.We appreciate for Reviewers’ warm work earnestly, and hope that the correction will meet with approval.

Once again, thank you very much for your comments and suggestions.

Yours sincerely,
Zuyun Luo
Corresponding author:
Name: Ruoyu Hong
E-mail: rhong@fzu.edu.cn

Reviewer 2 Report

The manuscript is written in a very simple way, the Englisgh has to be reviewed exahustivamnete by a native.

The results presented are interesting but to know the scope of the research the electrochemical properties of the electrodes prepared with PEGN/CB should be compared with those of electrodes prepared using another type of graphene, for example, reduced GO obtained by GO photoreduction.

Some mistakes should be corrected:

- Line 126: the coma after glass should be a dot.

- line 127-140: revised English.

- line 152: (e) should be before HRTEM.

- line 1l8: 2-d peak should be 2D.

- line 169: Figure 2b should be 3b.

- line 169-177: revised English.

- line 176: composition instead of information.

- line 180: The peaks correposnding to oxygen-containing .......

- The authors should coment the intensity increase of the peaks corresponding to aromatic C=C at 1622 cm-1 with increasing the time of plasma treatment.

- line 188: obtained instead of acquired.

- line 193: with plasma for 3 and 2 min, no peak......

-line 200: "unreduced" is not necessary, only GO.

- line 203: The coma before while should be a dot.

-line 204: add " as it can be observed in Figure 6c" after disappeared.

- line 202-20: revised English.

Author Response

Dear Reviewers: 

 Thank you for your reviewers’ comments concerning our manuscript entitled  “Plasma exfoliated graphene: preparation via rapid, mild thermal reduction of  graphene oxide and application in lithium batteries” (ID: materials-424271).  Those comments are all valuable and very helpful for revising and improving our paper, as well as the important guiding significance to our researches. We have studied  comments carefully and have made correction which we hope meet with approval.  Revised portion are marked in red in the paper. The main corrections in the paper and  the responds to the reviewer’s comments are as flowing: Responds to the reviewer’s comments:

Point 1: The manuscript is written in a very simple way, the English has to be reviewed exahustivamnete by a native.

Response 1: Thanks for the your kind advice. Native speakers have polished the article extremely careful to improve the expression of this manusript. Revised portion are marked in red in the revised manuscript.

Point 2: The results presented are interesting but to know the scope of the research the electrochemical properties of the electrodes prepared with PEGN/CB should be compared with those of electrodes prepared using another type of graphene. 

Response 2: By comparison with the privous research reference[37], adding a smaller amount of PEGN to form LiFePO4/graphene/carbon composite as a cathode material for lithium-ion batteries have the same result.

Point 3: Line 126: the coma after glass should be a dot. 

Response 3: The punctuation have been checked and corrected. 

Point 4: line 127-140: revised English.

Response 4: The Englisgh has be reviewed by a native.The revised portion are marked in red in the revised manuscript.

Point 5:line 152: (e) should be before HRTEM.

Response 5: We have made correction according to the Reviewer’s comments and marked in red in the revised manuscript.

Point 6: line 168: 2-d peak should be 2D.

Response 6: We are very sorry for our incorrect writing, the mistake have be corrected in the revised manusript and marked in red.

Point 7: line 169: Figure 2b should be 3b.

Response 7: We are very sorry for our negligence, the mistake have be corrected in the revised manusript and marked in red.

Point 8: line 169-177: revised English.

Response 8: We have re-written this part according to the Reviewer’s suggestion and marked in red.

Point 9: line 176: composition instead of information.

Response 9: As Reviewer suggested that We have used the composition instead the information in the revised manusript.

Point 10: line 180: The authors should comment the intensity increase of the peaks corresponding to aromatic C=C at 1622 cm-1 with increasing the time of plasma treatment.

Response 10: The reason of the intensity increase of the peaks corresponding to aromatic C=C at 1622 cm-1 with increasing the time of plasma treatment, primarily because the graphite structure is recovered with the increase of reduction degree.

Point 11: line 188: obtained instead of acquired.

Response 11: It is really true as Reviewer suggested that We have used the obtained instead the acquired in the revised manusript.

Point 12: line 193: with plasma for 3 and 2 min, no peak......

Response 12: From Figure 5, we can find that the product fabricated by plasma-treating GO for 2 and 3min, no peak was observed. So as Reviewer suggested that We have corrected the expression in the revised manuscript.

Point 13: line 200: "unreduced" is not necessary, only GO.

Response 13: As Reviewer suggested that We have deleted the “unreduced”, only used GO in the revised manusript.

Point 14: line 203: The coma before while should be a dot.

Response 14: The punctuation have been checked and corrected.

Point 15: line 204: add " as it can be observed in Figure 6c" after disappeared. 

Response 15: We have re-written this part according to the Reviewer’s suggestion in the revised manusript and marked in red.

Point 16: line 202-20: revised English.

Response 16: Considering the Reviewer’s suggestion We have re-written this part and marked in red in the revised manusript.

We tried our best to improve the manuscript and made some changes in the manuscript.  These changes will not influence the content and framework of the paper. And here we did not list the changes but marked in red in revised paper.We appreciate for Reviewers’ warm work earnestly, and hope that the correction will meet with approval.
Once again, thank you very much for your comments and suggestions.

Yours sincerely,
Zuyun Luo
Corresponding author:
Name: Ruoyu Hong
E-mail: rhong@fzu.edu.cn

Round  2

Reviewer 1 Report

The referee thanks the author for the response to the original comments. Despite this, the comments were not sufficiently answered.

In particular, regarding the graphene layer thickness, it is quite difficult for a single-layer graphene nanosheet to be stable in a chemical environment, especially in batteries-related nanomaterials. The tendency to aggregation of graphene has been extensively investigated in the batteries community, in fact in many cases, authors refer to “few layers graphene” . For this reason the author is strongly advised to thoroughly check the available literature papers on this matter.  

As concerns the original comment “The binder quotein the electrode preparation procedure is missing.”, there has been a misunderstanding due to a submission system typo, the request was about adding the binder quote (5%,10%, etc..), not “quotein” in the electrode slurry used for preparation.

In addition, the comparative analysis with a commercially available graphene,  is needed to obtain a good point of view on the actual performance of the material. In addition, a comparsion experiment with the PEGN as the only conductive additive should be added. The supplier of LiFePO4, Carbon Black, Binder and electrolyte solution should be added.

The referee invites the author to take the needed time to add the requested data, in order to improve the overall quality of the publication.

Author Response

Dear Reviewers:

Thank you very much for your comments on our manuscript entitled “Plasma exfoliated graphene: preparation via rapid, mild thermal reduction of graphene oxide and application in lithium batteries” (ID: materials-424271). These comments are very valuable and helpful for improving our manuscript.

We have read the comments carefully and have made correction according to your comments.

Point 1: In particular, regarding the graphene layer thickness, it is quite difficult for a single-layer graphene nanosheet to be stable in a chemical environment, especially in batteries-related nanomaterials. The tendency to aggregation of graphene has been extensively investigated in the batteries community, in fact in many cases, authors refer to “few layers graphene” . For this reason the author is strongly advised to thoroughly check the available literature papers on this matter.

Answer: Thank you for your kind suggestion. We strongly agree that the single-layer graphene is easy to aggregate. We read several referential papers, and some of them are listed  as references [30-32] in the revised manuscript. We now use “few layers graphene” instead of the single-layer graphene.

Point 2: As concerns the original comment “The binder quotein the electrode preparation procedure is missing.”, there has been a misunderstanding due to a submission system typo, the request was about adding the binder quote (5%,10%, etc..), not “quotein” in the electrode slurry used for preparation.

Answer: We are very sorry for our misunderstanding in the previous letter. When 10wt% CB is used as a conductive additive, the amount of binder is 10 wt %. When using PEGN (plasma exfoliated graphene)3 wt% and composite conductive agent of 2 wt% PEGN (plasma exfoliated graphene) and 1 wt% CB, the binder is added in an amount of 3 wt%. The corrected parts are marked in red in the revised manuscript.

Point 3: the comparative analysis with a commercially available graphene, is needed to obtain a good point of view on the actual performance of the material.

Answer: Currently, there are mainly two types of commercial graphene available. (1) Graphene was synthesised by oxidation-reduction method. There exist some problems in the produce process: Reduction process need to add the reducing agent, that will bring impurities such as ions; Reduction degree is not high and there are some oxygen-content groups on the surface of the graphene; The dispersion of the graphene is not good. These problems will affect the electrical properties of graphene. (2) Graphene was produced by exfoliation of natural graphite. There are many impurity in the natural graphite; It is necessary to add the solvents, surfactants and other additives to improve exfoliation efficiency and this will bring more impurities; The layer of graphene is usually much more than ten.

Point 4: a comparison experiment with the PEGN as the only conductive additive should be added.

Answer: Thank you for the suggestion.The comparison experiment with the PEGN as the only conductive additive was added in the revised manuscript. The results show that the performance of the composite conductive agent of 2 wt% PEGN and 1 wt% CB is better than the conductive agent of 10 wt% CB or 3 wt% PEGN, respectively. The main reason is that there is a good complementary effect in the contact modes of the graphene and carbon black when using the composite conductive agent. It can establish a more effective “long-range” and “short-range” conductive network inside the electrode. Besides, this binary conductive agent can improve the dispersion of graphene and prevent the aggregation of graphene, that further improve the efficiency of the electrical conduction.

Point 5: The supplier of LiFePO4, Carbon Black, Binder and electrolyte solution should be added.

Answer: Thank you for the suggestion. The LiFePO4 (Industrial purity) and N-methyl-2-pyrrolidone (Battery level) were supplied by Sinopharm Chemical Reagent Co., Ltd. The Carbon Black and Binder (Industrial purity) were obtained from Decatur Battery Technology Co., Ltd. The lithium foil and electrolyte solution (Battery level) come from Dongguan Shanshan Battery Materials Co., Ltd. These were added in the revised manuscript.

We tried our best to improve the manuscript and hope that the correction will meet with approval. We appreciate the Reviewers’ suggestion earnestly. Once again, thank you
very much for your comments and suggestions.

Yours sincerely,
Ruoyu Hong, Ph.D.

Professor at Cheg, Fuzhou Univ.
E-mail: rhong@fzu.edu.cn

Reviewer 2 Report

The authors have addressed properly the reviewer suggestions. 

Author Response

Dear Reviewers:

Thank you very much for your comments on our manuscript entitled “Plasma exfoliated graphene: preparation via rapid, mild thermal reduction of graphene oxide and application in lithium batteries” (ID: materials-424271). These comments are very valuable and helpful for improving our manuscript.

Responds to the reviewer’s comments:

Point 1: The authors have addressed properly the reviewer suggestions.

Answer:: Thank you very much for your recognition and support of our work.

We tried our best to improve the manuscript and hope that the correction will meet with approval. We appreciate the Reviewers’ suggestion earnestly. Once again, thank you
very much for your comments and suggestions.

Yours sincerely,
Ruoyu Hong, Ph.D.

Professor at Cheg, Fuzhou Univ.
E-mail: rhong@fzu.edu.cn